# Graph of Circuits with GNN for Exploring the Optimal Design Space

**Aditya Hemant Shahane**[1]     **Swapna Manjiri**[2]     **Ankesh Jain**[2]     **Sandeep Kumar**[1,2,3]

[1]Bharti School of Telecommunication Technology and Management     [2]Electrical Engineering

[3] Yardi School of Artificial Intelligence

Indian Institute of Technology Delhi

{bsy217530,een212025,ankesh,ksandeep}@iitd.ac.in

## Abstract

The design automation of analog circuits poses significant challenges in terms of the large design space, complex interdependencies between circuit specifications, and resource-intensive simulations. To address these challenges, this paper presents an innovative framework called the Graph of Circuits Explorer (GCX). Leveraging graph structure learning along with graph neural networks, GCX enables the creation of a surrogate model that facilitates efficient exploration of the optimal design space within a semi-supervised learning framework which reduces the need for large labeled datasets. The proposed approach comprises three key stages. First, we learn the geometric representation of circuits and enrich it with technology information to create a comprehensive feature vector. Subsequently, integrating feature-based graph learning with few-shot and zero-shot learning enhances the generalizability in predictions for unseen circuits. Finally, we introduce two algorithms namely, EASCO and ASTROG which upon integration with GCX optimize the available samples to yield the optimal circuit configuration meeting the designer's criteria. The effectiveness of the proposed approach is demonstrated through simulated performance evaluation of various circuits, using derived parameters in 180 nm CMOS technology. Furthermore, the generalizability of the approach is extended to higher-order topologies and different technology nodes such as 65 nm and 45 nm CMOS process nodes.

## 1   Introduction

Designing Electronic Design Automation (EDA) tools for analog circuits is challenging as it involves optimizing multiple parameters and their non-linear dependencies. Machine learning models are increasingly being integrated within EDA tools aiming to make this process flexible and suited for analog design across different circuits and technology nodes. Instead of simulation-based global optimization [14], which can be time-consuming for stricter requirements [8], a surrogate-model-assisted optimization approach using less computationally expensive approximation models have been proposed.

Regarding surrogate models, Bayesian optimization and Gaussian process models have been more effective than standard artificial neural networks and radial basis functions. However, Gaussian process modeling is computationally expensive [3], and as circuit complexity increases, inter-sample relations become harder to capture, training time increases and generalization ability is lost. To address these challenges, graph-based models that integrate Graph neural networks (GNNs) [5, 22, 17, 29] have emerged as a powerful technique in high-level synthesis, behavior modeling [9], placement and routing [20, 18, 28, 13], analog and mixed-signal layout exploration [25], making them a plausible choice for graph-based surrogate models. Graphs capture both topological and feature-based information, resulting in faster modeling and optimization of complex systems. Despite their

success, graph-based methods still require improvement in terms of independence from labeled data and generalization ability across different circuit topologies and technology nodes. For optimization algorithms, previous works have employed reinforcement learning [26], and evolutionary algorithms [6]. However, the models or policies that are learned via these methods are highly tailored to the environment or the objective of the optimization and lack reproducibility. Another popular method is the particle swarm intelligence [23] however, it easily falls into local optima in the high dimensional space and suffers from low convergence rate.

This paper introduces a novel and innovative approach to optimize circuit design by leveraging graph representation and graph-based semi-supervised learning. The approach is designed to enhance accuracy and efficiency while reducing the reliance on extensive labeled datasets. To achieve this goal, a semi-supervised learning framework is employed for the graph-based surrogate model. The accuracy of the approach is compared against the benchmark model ParaGraph [22], and its superior performance is demonstrated. Additionally, a new method is introduced to create a comprehensive feature vector that integrates information about various technology nodes and topologies, emphasizing generalizability in zero-shot and few-shot learning frameworks. By integrating these approaches, two new optimization methods on graph-based surrogate models are proposed: Efficient Analog Sizing via Constrained Optimization (EASCO) and Analog Sizing through Real-time Online Graphs (ASTROG). The paper is structured into seven sections, covering graphs, graph-based techniques, optimization strategies, the proposed approach, simulation results, and concluding remarks.

## 2 Related work

Construction of a graph-based surrogate model starts with circuit representation as graphs and the creation of a labeled graph dataset. The task is formulated as a graph-level regression problem with each graph mapped to its corresponding label vector (performance metric) using Graph Neural Networks (GNNs).

**Graph representation for Analog and Mixed-signal circuits** A graph is a data structure used to store semantic information with the help of its unique structure formed using nodes and edges. A graph with features is denoted by $G = (V, E, W, X, Y)$ where $V = \{v_1, v_2, \ldots, v_n\}$ is the vertex set, $E \subseteq V \times V$ is the edge set and $W$ is the adjacency (weight) matrix. We consider a simple undirected graph without self-loop: $W_{ij} > 0$, if $(i,j) \in E$ and $W_{ij} = 0$, if $(i,j) \notin E$. Finally, $X \in \mathbb{R}^{n \times d} = [X_1, X_2, ..., X_n]^T$ and $Y \in \mathbb{R}^{n \times m} = [Y_1, Y_2, ..., Y_n]^T$ are the feature matrix and label matrix respectively, where the row vectors $X_i \in \mathbb{R}^d$ and $Y_i \in \mathbb{R}^m$ are the feature vector and label vector, respectively associated with one of $n$ nodes of the graph $G$. The most common approach

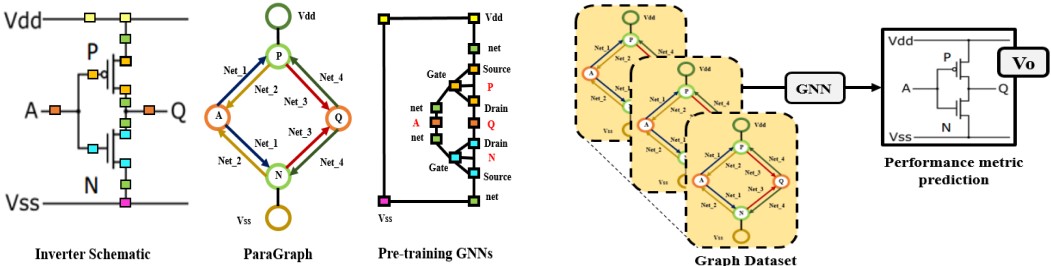

Figure 1: The figure (a) depicts two distinct methods for representing circuits as a graph. In [22] circuit components are depicted as nodes while nets carry edge-based information. To this, [5] suggested explicitly representing device terminals as nodes to avoid ambiguity when the device has identical net connections between two or more terminals. Meanwhile, in figure (b), we can see the training framework of ParaGraph [22] method. For the example circuit used, multiple graphs and corresponding label vectors are used for training, and prediction on the unlabeled graph is performed.

for creating a circuit graph is based on the circuit's topological design. In this approach, circuit components such as resistors, capacitors, and transistors are depicted as nodes, while the circuit wiring (net) is mapped using edge-based information that connects different nodes [22]. This approach results in a heterogeneous graph that is useful for analyzing and developing complex electronic systems. As a result, a graph-level prediction problem [5, 22, 17] arises where we have a set of $n$ graphs denoted by $\{G_1, G_2, \ldots, G_n\}$, each with its own set of nodes and edges, a function $f$ which

maps these graphs to corresponding prediction metric ($Y_i$), where $Y_i = f(G_i)$. Let $X_i$ be the feature matrix for $i^{th}$ graph where $X_i \in R^{n \times F}$ where $n$ is the number of nodes in a single graph and $F$ is the features of each node. Overall, the problem of graph-level prediction using GNNs involves learning the function $f$ based on graph structure and node features that map the input graphs to the desired prediction metric. Building upon the previously presented method for representing circuits as a graph, [5] proposed a refined approach by explicitly including device terminals as nodes in the graph. This modification results in featureless edges.

While this approach offers a detailed understanding of the interconnection between different circuit components, it suffers from several limitations: (i) As the circuit size increases, the graph becomes densely interconnected, leading to a large number of nodes and edges, making it challenging to comprehend the overall behavior of the circuit. (ii) Creating a large labeled graph dataset is time-consuming and computationally expensive, which hinders effective GNN training. (iii) This framework lacks inter-graph connections, which reduces its reliability. (iv) Since graph regression problems cannot be applied to semi-supervised frameworks, the reliance on labeled datasets is inevitable. (v) Additionally, training and validating over multiple graphs ($n$) with $N$ nodes and $E$ edges when passed through $L$ layer GNN makes the process slow, and the high time-complexity ($O(n * LN^2 E)$) and space-complexity ($O(n * N^2 L)$) further adds to the computational burden. (vi) Finally, transferring knowledge across different topologies becomes challenging due to varying graph sizes.

## 3    Proposed Formulation and Background

For our case, consider $n$ number of circuit instances indexed as $i = 1, \ldots, n$, where $X_i \in \mathbb{R}^d$ be the feature vector and $Y_i \in \mathbb{R}^m$ the label vector corresponding to $i$-th circuit. A feature vector $X_i$ contains the circuit-level parameters e.g. resistance, capacitance, transistor widths, lengths, junction voltages etc and the corresponding label vector $Y_i$ contains performance parameters, including *Gain, Bandwidth, Noise, etc*. The problem under consideration is:

*Given $n = l + u$ circuit instances and their parameters $X = (X_l, X_u)$ along with labels of only $l$ number of circuit instances as $Y_l$, how can we predict the labels of the remaining $u$ number of circuit instances $Y_u$?*

The circuit performance parameters or the labels $Y_i$ are generally obtained using expensive SPICE simulation, thus having a large set of labeled data is a luxury. Limited labeled data where $l < u$ is a significant hurdle in learning models for EDA. Graph-based Semi-supervised learning attempts to address this by learning from both labeled as well as unlabeled data.

Consider sets $T_l = \{(X_i, Y_i)\}_{i=1}^{l} \in \mathcal{X} \times \mathcal{Y}$ and $T_u = \{(X_i)\}_{i=l+1}^{l+u} \in \mathcal{X}$, where $\mathcal{X}$ is the domain set and $\mathcal{Y}$ is the label set. Suppose the elements of the set $T_l$ be sampled from probability distribution $\mathbb{P}$ over the ordered pair $\mathcal{X} \times \mathcal{Y}$. Samples of the domain set $X_i \in \mathcal{X}$ will follow the marginal distribution $\mathbb{P}_\mathcal{X}$ of $\mathbb{P}$. For the limited labeled data scenario, the goal of a learning algorithm is to estimate conditional distribution $\mathbb{P}(Y|X)$ using both the labeled and unlabeled data as the training data. To incorporate the additional information of $\mathbb{P}_\mathcal{X}$ obtained because of the unlabeled data by capturing the intrinsic geometry of the marginal distribution through the graph of both unlabeled and labeled data and performing label propagation and feature aggregation over the neighborhood. The guiding principle is that if two data points from the domain set are close in the intrinsic distribution $\mathbb{P}_\mathcal{X}$ then the corresponding conditional distributions should be similar. For the circuit prediction, it means that given two circuit instances with similar features $X_i \sim X_j$ then the corresponding label vectors should also be similar $Y_i \sim Y_j$.

To build a learning framework based on the above guiding principle, these are the two main steps i) First use the feature space to obtain a geometric representation such that the samples having similar features are represented closer and connected ii) Build a learning model such that geometric information be leveraged for learning the labels of unknown samples.

### 3.1    Proposed Framework for Semi-Supervised Learning on Graph of Circuits

We introduce a graph-based semi-supervised learning technique for building a surrogate model (GCX) for analog circuit simulation with limited labels. The proposed framework consists of four main stages: i) Firstly for obtaining feature-aware geometrical representation we construct a graph of circuits using circuit features $X = (X_l, X_u)$ in the form of an adjacency matrix $A$, where each circuit

$i$ is a node of the graph and the circuit-level parameters, such as resistance, capacitance, transistor widths and lengths, and junction voltages, as its corresponding feature vector $X_i \in \mathbb{R}^k$. ii) Further using the graph adjacency matrix, feature matrix, and available set of circuit performance matrix together, i.e., $G(X, A, Y_l)$ with the graph neural network pipeline to build a machine learning model, iii) Constructing a comprehensive feature vector that facilitates generalizability(discussed later), iv) predicting the performance parameters $Y_u$ corresponding to $X'_u$s. Fig 2 illustrates the outline of the proposed framework:

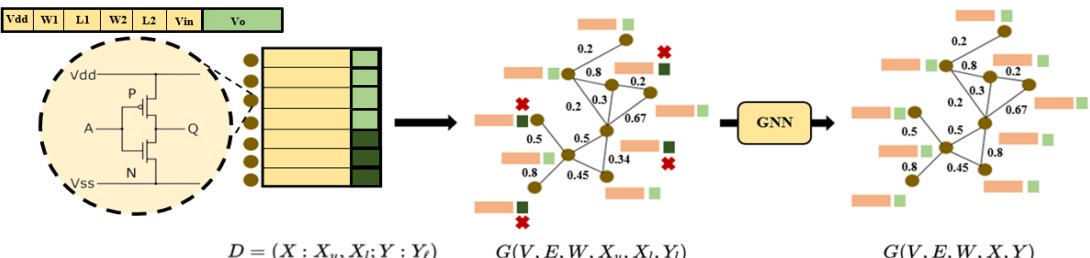

$$D = (X : X_u, X_l; Y : Y_\ell) \qquad G(V, E, W, X_u, X_l, Y_l) \qquad G(V, E, W, X, Y)$$

Figure 2: **Graph of Circuits Explorer (GCX)**: The original dataset (D) contains both labeled ($X_l$,$Y_l$) and unlabeled ($X_u$) samples, each circuit is a node of the graph and sample as the feature vector of the associated node that will serve as nodes in the graph; the graph is learned (weighted adjacency-W) using the suggested formulation; and the learned graph is passed through a GNN model for label prediction for unlabeled nodes. This forms a **GCX(GNN)** surrogate model. For the above circuit $X_{li} = (V_{dd}, W1, L1, W2, L2, V_{in})$ and $Y_{li} = Vo$.

---

**Algorithm 1 GCX(GNN)**

---

**Input:** Feature matrix ($X_l$, $X_u$), label matrix ($Y_l$)
**Output:** Optimal design ($X_i^*$)

1. $G(V, E, W) \longleftarrow$ Learning Graph of Circuits Using ($X_l, X_u$)
2. $Y_u \longleftarrow$ Label prediction of unlabeled circuits using GNN with $G(V, E, W, X_l, X_u, Y_l)$;
3. GCX(GNN) $\longleftarrow G(V, E, W, X_u, X_l, Y_l)$ + GNN; surrogate model
4. $X_i^* \longleftarrow$ GCX(GNN) + EASCO/ASTROG

---

Algorithm 1, which we denote by GCX(GNN), summarizes the outline of the proposed framework. The implementation details of each step are discussed subsequently.

This approach allows us to incorporate multiple circuit instances into a single graph while accommodating different technology nodes and topological diversity. We achieve this while maintaining a homogeneous graph with node feature vectors of equal lengths, making the training of GNNs in a semi-supervised framework efficient and cost-effective. Compared to the previous approach that handles circuits as graphs, our approach with $N$ nodes in a graph with $E$ edges when passes through $L$ layer GNN has an overall time complexity of $O(L * N^2 E)$ and space complexity of $O(N^2 L)$.

### 3.2  How to Learn Graph of Circuits to Encode Circuit Similarity

Consider the feature matrix of circuit instances $X = [X_1, X_2, \ldots, X_n]^T$ where $X_i$ is the feature of $i^{th}$ circuit which corresponds to $i^{th}$ node of a graph. In the context of modeling signals or features with graphs, the widely used assumption is that the signal residing on the graph changes smoothly between connected nodes [10]. For example, if two circuit instances $X_i$ and $X_j$ have similar features then their Euclidean distance $\|X_i - X_j\|^2$ should be smaller but connecting edge weight which quantifies similarity i.e. $w_{ij}$ should be large. The Dirichlet energy (DE) is used for quantifying the smoothness of the graph signals which is defined as:

$$\text{DE} = \frac{1}{2} \sum_{i,j} w_{ij} \|X_i - X_j\|^2$$

The lower value of Dirichlet energy indicates a desirable configuration [31, 10]. Smooth graph signal methods are an extremely popular family of approaches for a variety of applications in machine learning and related domains [12, 10]. In order to build a graph of circuits we will use the smoothness

of signals on graph assumption. Nodes connected with stronger weights indicate similar features and performance indices, facilitating knowledge transfer to new circuit instances. Finally, we learn a weighted adjacency matrix $W$ by minimizing the Dirichlet energy combined with other sparsity regularization terms which entails solving the following optimization problem:

$$w := \arg\min_{w \in w_m} \frac{1}{2} \sum_{i,j} w_{ij} \|X_i - X_j\|^2 - \alpha 1^\top \log(Sw) + \beta \|w\|_2^2$$

where $\alpha > 0$ and $\beta \geq 0$ controlling the properties of the resultant graph, $\|w\|_2^2$ ensures the graph is sparse, while $1^\top \log(Sw)$ ensures that every node will have some connections. This problem is a convex optimization problem and efficiently solvable [12, 10].

### 3.3 Comprehensive feature vector representation for GCX

To streamline GNN training across various technology nodes and topologies, we extend the initial feature vector with one-hot encoded bits, specifically allocated for the technology file and topology. This augmentation generates a new feature vector that facilitates relevant pattern learning by the GNN. This approach significantly reduces computational costs, making it more viable for practical applications. With this trained GNN model, it becomes possible to accurately predict the performance of circuits that are yet to be seen, thereby enhancing circuit design and optimization. Further information about our proposed method is available in Figure 3.

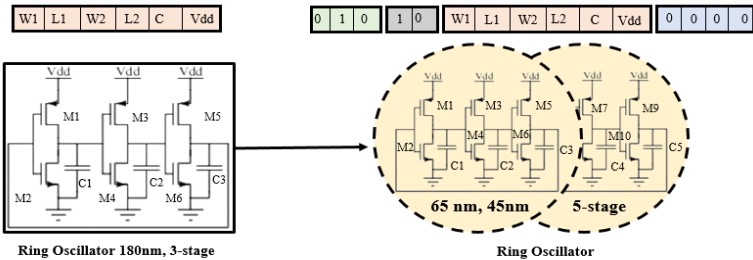

Figure 3: **Knowledge Transfer**: To enhance the feature vector, we append it with one-hot encoded technology files and circuit topology details, resulting in a new feature vector. The technology files used, 180nm, 65nm, and 45nm, are represented using 3 bits. To represent the original and higher-order topology, 2 bits are used, with an additional $k$ bits assigned for circuit element features. The first figure shows the ring oscillator implemented with the 65nm technology file, while the second figure displays the 5-stage topology of the circuit.

### 3.4 Graph Circuit Explorer-Graph Neural Networks: GCX(GNN) surrogate model

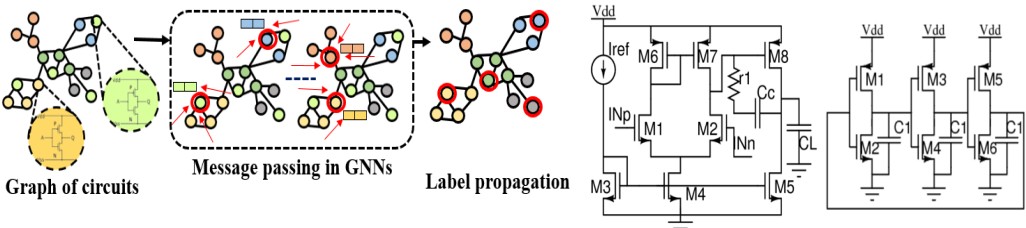

Figure 4: The figure (a) depicts message passing in GNNs: Graph is passed through the GNN architecture; nodes interact with each other via message passing and feature aggregation for target node is carried out. Unlabeled nodes gather information from its labeled neighborhood in form of embeddings. Embeddings updated over $L$ layers is used for label prediction. To better understand GNN architectures, refer: [11, 7, 27]. Meanwhile, in figure (b), we can see schematics of (i) Two-stage OTA with Miller Compensation and (ii) Three stage Ring Oscillator on which the algorithms will be implemented.

After learning a graph using the feature vector description provided, the graph undergoes label propagation through the GNN architecture. In this process, each node represents a circuit instance and the graph consists of both labeled and unlabeled nodes. During each layer of the GNN, the nodes interact through message passing, and feature aggregation is carried out. The unlabeled circuit instance acquires information from its labeled neighbors in the form of embeddings. The embeddings

are iteratively updated across the $L$ layers of the GNN, ultimately leading to the prediction of a label for the unlabeled circuit instance (node). The formulation for the embedding update can be understood from the equation given below. The process can be understood from Fig:4

$$h_u^{(k)} = \sigma \left( W_{\text{self}}^{(k)} h_u^{(k-1)} + W_{\text{neigh}}^{(k)} \sum_{v \in N_u} h_v^{(k-1)} + b^{(k)} \right)$$

$h_u^{(k-1)} \in \mathbb{R}^{d^{(k-1)}}$ : Node embeddings of target node $u$ at layer $k$. $W_{\text{self}}^{(k)}, W_{\text{neigh}}^{(k)} \in \mathbb{R}^{d^{(k)} \times d^{(k-1)}}$: are the learnable parameters, $b^{(k)} \in \mathbb{R}^{d^{(k)}}$ : represents the bias term and $\sigma$: represents elementwise non-linearity (e.g., a tanh or ReLU). GCX, the approach of creating a graph of circuits, seamlessly integrates with any GNN architecture, showcasing its flexibility across circuits and architectures. Extensive experiments provide conclusive evidence of its effectiveness.

## 4   Efficient Analog Sizing via Constrained Optimization (GCX-EASCO)

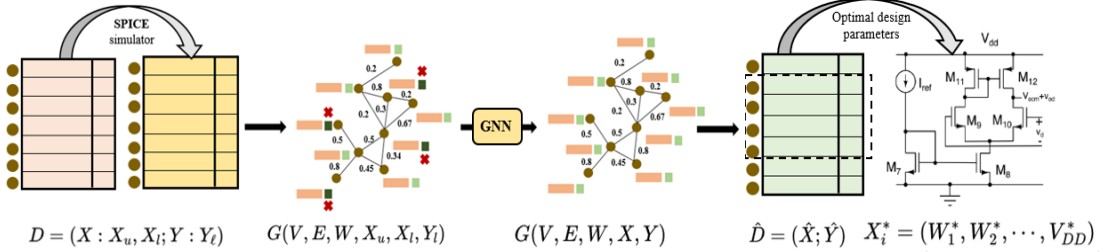

Figure 5: **GCX-EASCO algorithm**: Initially points are uniformly sampled and labels are generated based on the value of $l$; graph $G(V, E, W, X_u, X_l, Y_l)$ is learned using all the points based on features; GNNs in a semi-supervised framework learn the embeddings corresponding to labeled samples, which are used for label prediction corresponding to unlabeled nodes $G(V, E, W, X, Y)$. Ultimately, constrained optimization over the subset $\hat{D}$ of the original dataset gives optimum parameters $X_i^*$.

Our new algorithm employs a more efficient graph-based surrogate model to obtain optimal circuit parameters. The algorithm uses a static, offline training framework that has been recommended in [15, 16]. The algorithmic flow is described in Fig:5 and the following is the step-by-step overview:

The EASCO algorithm consists of four main components. Firstly, in the data generation phase, $n$ points are uniformly sampled from a $d$-dimensional feature space such that $X_i \in [a_i, b_i]$ where $a_i$ and $b_i$ are the lower bound (LB) and upper bound (UB) for each feature. This creates a dataset (D) such that $D \in \mathbb{R}^{n \times d}$. Next, a graph is learned (as previously discussed) using all the $n$ sampled points, with a subset of these points simulated under different labeled data settings ($l$), such as 30%, and 50%, by adopting a semi-supervised learning framework. The labeled points, $X_{train} \in \mathbb{R}^{l \times d}$ are used for training, while the rest, $X_{test} \in \mathbb{R}^{(n-l) \times d}$ are reserved for testing. In the third component, GNN models [11, 7, 27] are used to make predictions. Lastly, in the constrained optimization component, uniform sampling and integration with label propagation over unlabeled samples offer an overall comprehension of the feature(search) space. With constraints in place, the process returns a subset of the original search space $\hat{D} \in \mathbb{R}^{k \times d}$ where $k << n$. The objective function is optimized using Differential Evolution (DE) algorithm [4] applied to the resulting parameter space, yielding the most optimal parameters.

## 5   Analog Sizing Through Real-time Online Graphs (GCX-ASTROG)

The previous algorithm efficiently minimized simulations once coarse boundaries were established. However, handling an increasing number of specifications made it challenging to identify these boundaries. To overcome this, we developed an online training version of the algorithm that requires minimal initial data. It utilizes an optimization algorithm, a surrogate model, and an infill sampling criterion [21] to guide the search towards global optima. The algorithm updates iteratively and provides the optimal circuit design parameters upon meeting the stopping criterion. Our adoption of the ranking criterion from [2] ensures consideration of all constraints and enables a precise approach

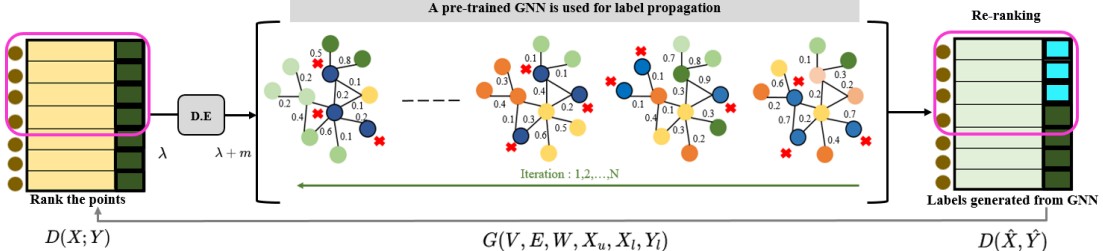

Figure 6: **GCX-ASTROG algorithm**: The process starts with sampling and labeling a small set of points ($\alpha$). A novel sampling criterion guides the algorithm towards a specific feature space region. The top $\lambda$ points create a population (P) for the DE algorithm, generating $m$ unlabeled points at every iteration of ASTROG. A semi-supervised learning framework uses $\lambda + m$ points to learn a graph $G(V, E, W, X_u, X_l, Y_l)$. A pre-trained GNN propagates labels over the $m$ points at each iteration. The best solution is integrated into the original dataset, and the process continues until the iteration budget is met.

to achieving optimal design outcomes. See Fig. 6 for the algorithm flow, and the step-by-step overview is as follows:

Initially, a dataset ($D$) is generated with $\alpha$ samples such that $D \in \mathbb{R}^{\alpha \times d}$. The initial dataset is created using SPICE simulations within the bounds, denoted as $a_i$ and $b_i$ for each design metric. Stopping criteria are predefined, and the optimal design is outputted if these criteria are met; otherwise, the process moves to the next step. In step three, all designs in the database are ranked using the adopted infill sampling criterion [2], and the $\lambda$ best designs are selected. New bounds $\tilde{X}_i = [\tilde{a}_i, \tilde{b}_i]$ are obtained based on these designs, where $\tilde{X}_i \in \mathbb{R}^{\lambda \times d}$. The Differential Evolution (DE) algorithm [4] is used to optimize the objective function by generating $m$ child solutions from the population within the newly obtained bounds. Next, a graph is learned (with the previously discussed approach) by combining the features of best $\lambda$ labeled points with the $m$ unlabeled points obtained from the DE algorithm. The obtained graph is passed through the pre-trained GNN models and labels are predicted for the $m$ unlabeled nodes. After the label prediction, the $\lambda + m$ labeled points are re-ranked using the infill sampling criterion [2]. The optimal design and its corresponding performance metric (label) are then selected from the $\lambda + m$ labeled points and incorporated into the original database. This process is repeated until the predetermined iteration budget for each circuit is exhausted.

## 6 Experiments

In this section, we showcase the experimental results of our two proposed algorithms: EASCO and ASTROG. To evaluate the effectiveness of these algorithms, we chose Two-stage OTA with Miller Compensation which is operating mostly in a linear region, and the Ring Oscillator (Three-stage) which is operating mostly in a nonlinear region as our test cases. The Miller Compensated OTA has 11 design variables, while the Ring Oscillator has 6 design variables. The performance evaluation of our algorithms was conducted using multiple metrics, which will be discussed in detail later. Our experimental workflow comprises four stages. We initiate with a comprehensive description of dataset generation using SPICE simulation and subsequently provide an exhaustive analysis of the algorithm's operational efficiency.

### 6.1 Harnessing EDA tools- SPICE simulator for Dataset generation

We have considered two test cases belonging to different families of circuits: **Amplifiers** and **Oscillators**. Design space bounds, Performance metrics and Specifications are carefully considered after designer's recommendation (mentioned subsequently). Features (design space points) $X = (X_u, X_l)$ are generated by uniformly sampling points across the $d$-dimensional space forming a matrix of size $X \in \mathbb{R}^{n \times d}$. Since we employ semi-supervised learning we simulate labels corresponding to only $l$ samples where $l << n$ (samples are randomly chosen to avoid bias). The dataset obtained is : $(X_l, X_u, Y_l)$.

**Two-stage Miller Compensated OTA**: Design parameters ($X_i$) are: Reference current ($I_{ref}$), Reference Voltage ($V_{dd}$), Transistor widths and length ($W_{1-8}, L_{1-8}$), Capacitance ($C_c, C_L$) and load resistance ($r_L$). Performance metrics ($Y_i$) are : Gain, Unity Gain Bandwidth (UGB), Gain Margin (GM), Phase Margin (PM), Noise and Power.

**Three-stage Ring Oscillator**: Design parameters $(X_i)$ are: Reference Voltage $(V_{dd})$, Transistor widths and length $(W_{1,2}, L_{1,2})$ and Capacitance $(C_1)$. Performance metrics $(Y_i)$ are: Frequency, rms Jitter, Delay and Power.

## 6.2 Performance evaluation of GCX surrogate model

To evaluate the efficacy of our proposed approach of learning graph of circuits, we conducted tests by integrating it with various GNN architectures [11, 7, 27]. For each test case we learned a graph with $n$ = 1k nodes and training was performed under different labeled data settings, with $l$ being set to 30% and 50%. We compared the results of our proposed node-level regression approach with benchmark method ParaGraph [22], which is a graph-level regression setting, and ESSAB [2], a novel two-layer neural network for regression. The methods used for comparison do not facilitate semi-supervised learning hence experimental setup was adopted as in mentioned in [22, 2].

We implemented the graph formation technique described in ParaGraph [22] for our selected test cases. We established a graph level regression problem consisting of 15k graphs along with their corresponding label vectors. Additionally, we constructed a GNN architecture as outlined in the paper. However, during experimentation, we observed that the resulting $R^2$ scores were predominantly negative. Consequently, we opted not to showcase the results for the surrogate model. We attribute this failure to two main factors: (i) The scarcity of node and edge features specific to the circuit under study. The ParaGraph [22] method was initially designed for industrial circuits with longer feature vectors, which might have impacted the performance on our dataset. (ii) The limited number of graphs available for training. Although the training loss initially decreased, it quickly reached saturation, leading to ambiguous predictions. Considering these factors, we believe that a more comprehensive feature set and a larger training dataset would be necessary to achieve more accurate results with the proposed approach.

We utilized the ESSAB surrogate model [2] with a dataset of 1k labeled samples. To ensure consistency in training conditions, we maintained a 50% training-to-testing ratio, similar to our approach.

| Miller Compensated Two-stage OTA ($R^2$ scores) | | | | | | | | | | | | |
|---|---|---|---|---|---|---|---|---|---|---|---|---|
| GCX(.) | Gain (dB) | | UGB (MHz) | | GM (dB) | | PM (deg) | | Noise ($\mu V$) | | Power ($\mu W$) | |
| | $l$=30% | $l$=50% | $l$=30% | $l$=50% | $l$=30% | $l$=50% | $l$=30% | $l$=50% | $l$=30% | $l$=50% | $l$=30% | $l$=50% |
| (GCN) | 0.17 | 0.22 | 0.36 | 0.43 | 0.52 | 0.81 | 0.70 | 0.81 | 0.35 | 0.64 | 0.42 | 0.61 |
| (SAGE) | 0.42 | **0.60** | 0.43 | **0.60** | 0.89 | **0.91** | 0.93 | **0.94** | 0.96 | **0.97** | 0.97 | **0.99** |
| (GAT) | 0.19 | 0.41 | 0.33 | 0.41 | 0.89 | 0.91 | 0.79 | 0.85 | 0.41 | 0.61 | 0.60 | 0.71 |

Table 1: $R^2$ scores with different GNN architectures (Included value is the mean, averaged across 5 runs)

| Three-stage Ring Oscillator ($R^2$ scores) | | | | | |
|---|---|---|---|---|---|
| GCX(.) | Frequency (MHz) | | Delay ($\mu S$) | | Power ($\mu W$) | |
| | $l$=30% | $l$=50% | $l$=30% | $l$=50% | $l$=30% | $l$=50% |
| (GCN) | 0.30 | 0.44 | 0.28 | 0.55 | 0.35 | 0.63 |
| (SAGE) | 0.45 | **0.83** | 0.56 | **0.82** | 0.84 | **0.91** |
| (GAT) | 0.41 | 0.50 | 0.34 | 0.58 | 0.36 | 0.58 |

Table 2: $R^2$ scores with different GNN architectures (Included value is the mean, averaged across 5 runs)

| Model | Gain | UGB | GM | PM | Noise | Power | Frequency | Delay | Power |
|---|---|---|---|---|---|---|---|---|---|
| **GCX(SAGE)** | **0.60** | **0.60** | **0.91** | **0.94** | **0.97** | **0.99** | 0.83 | **0.82** | **0.91** |
| ESSAB [2] | 0.57 | 0.50 | 0.81 | 0.93 | 0.89 | 0.89 | **0.89** | 0.71 | 0.88 |

Table 3: $R^2$ scores comparison between our best model GCX(SAGE) and ESSAB [2] surrogate model.

Our proposed approach, based on the semi-supervised learning framework, outperforms competing methods even when provided with minimal labeled data. We demonstrate the flexibility of our approach by evaluating it on two test cases representing different classes of circuits. Furthermore, we obtain a highly accurate graph-based surrogate model that will be utilized in the subsequent algorithms proposed in this work. Results of our approach are presented in Tables:1,2,3 respectively.

## 6.3 Comparative analysis between different Optimization strategies

Our approach involves first obtaining a highly accurate surrogate model, which we subsequently employ in two algorithms, as discussed earlier, to determine the optimal parameters for the given design problem. To evaluate the efficacy of our proposed algorithms, we compare them against two benchmark optimization methods - the DE algorithm [24] and the BO-EI algorithm proposed in [1]. To ensure reproducibility, we run each algorithm ten times. We then solve a single objective constrained optimization problem, based on the designer's recommendations, for each circuit, and report the obtained results. We selected the Figure of Merit (FOM) as the objective function to be minimized in our study. To define the FOM, we extensively reviewed the literature [19, 30] and derived a customized expression that aligns with our specific requirements. Further details regarding the formulation of this FOM can be found in the **supplementary materials**. The FOM expression we developed effectively balances the trade-off between different specifications, enabling us to optimize the input parameters for the given test cases. The best-case results are depicted in **blue**, while the second-best results are highlighted in **black**.

| Miller Compensated Two-stage OTA (Optimization) | | | | | | | | | | | | | |
|---|---|---|---|---|---|---|---|---|---|---|---|---|---|
| Objective: Minimize FOM st. Gain > 50 dB ; UGB > 100 Mhz ; PM > 45 deg ; GM > 15 dB ; Power < 900 $\mu$W ; Noise < 600 nVrms | | | | | | | | | | | | | |
| Model | Gain (dB) | | UGB (MHz) | | GM (dB) | | PM (deg) | | Noise ($nV$) | | Power ($\mu W$) | | Success |
| (specs) | Max | Min | Max | Min | Max | Min | Max | Min | Max | Min | Max | Min | (out of 10) |
| DE [24] | 58.6 | 53.3 | 190 | 165 | 19.2 | 17.8 | 46.7 | 45.0 | 478 | 463 | 815 | 633 | 04/10 |
| BO-EI[1] | **68.3** | 65.6 | **225** | 190 | 18.9 | 17.8 | **52.0** | 45.6 | 470 | 462 | 627 | 505 | 03/10 |
| **EASCO** | 67.3 | 64.4 | **215** | 192 | **19.2** | 18.5 | 47.2 | 45.5 | 470 | **462** | 722 | **490** | **04/10** |
| **ASTROG** | **80** | 61.6 | 145 | 106 | **26** | 23.5 | **52.5** | 47.9 | 497 | **446** | 716 | **318** | **06/10** |

Table 4: Comparison table for different optimization strategies [24, 1] with EASCO and ASTROG

ASTROG stands out as the undisputed champion among all competing algorithms. The convergence speed achieved by ASTROG is approximately **8x** faster compared to the BO-EI algorithm [1]. The results are as shown in Table:4.

| Three-stage Ring Oscillator (Optimization) | | | | | | | | | |
|---|---|---|---|---|---|---|---|---|---|
| Objective: Minimize FOM st. Freq > 1000 MHz ; Jitter < 10 pS ; Delay < 200 pS ; Power < 600 $\mu$W | | | | | | | | | |
| Model | Frequency (MHz) | | Jitter($pS$) | | Delay($pS$) | | Power($\mu W$) | | Success |
| (specs) | Max | Min | Max | Min | Max | Min | Max | Min | (out of 10) |
| DE [24] | 1265 | 1005 | 5.07 | 0.99 | 165 | 131 | 302 | 256 | 02/10 |
| BO-EI[1] | **2103** | 1047 | 2.73 | $10^{-7}$ | 159 | **79** | 407 | **66.4** | **06/10** |
| **EASCO** | 1444 | 1396 | 1.7 | $10^{-7}$ | 119 | 115 | 360 | 331 | **09/10** |
| **ASTROG** | **2033** | 1175 | 6.05 | **0.2** | 141 | **81.9** | 502 | **182** | 04/10 |

Table 5: Comparison table for different optimization strategies [24, 1] with EASCO and ASTROG

Among the competing algorithms, BO-EI [1] proved to be a formidable rival for our proposed algorithms. Although EASCO maintained its higher rank in terms of reproducibility, ASTROG delivered the second best results. However, the key factor in determining the champion algorithm lies in the speed of convergence towards the optimal value. In this regard, ASTROG's impressive **4x** faster convergence clearly establishes it as the ultimate winner. The results are as shown in Table:5. For a comprehensive analysis of the timing results for each algorithm, please refer to the **supplementary materials**.

## 6.4 Knowledge Transfer using Zero-shot and Few-shot learning

To broaden the scope of our approach, we incorporated different technology nodes and topologies into our study. We selected a Three-stage ring oscillator as our initial test case and implemented the procedure for creating a comprehensive feature vector. We specifically focused on conducting few-shot (FS) learning experiments using 180 nm technology nodes. We then assessed the generalization of the trained model to 65 nm technology nodes and a five-stage topology. Additionally, we performed zero-shot (ZS) learning experiments using 45 nm technology nodes. This allowed us to evaluate the model's performance without any prior training on the specific technology node.

| Ring Oscillator (Three and Five-stage, 180 nm, 65 nm, 45 nm) | | | | |
|---|---|---|---|---|
| Model | 3-S,FS (65 nm) | 3-S,ZS (45 nm) | 3to5-S,FS (65 nm) | 3to5-S,ZS (45 nm) |
| | $l$=30% | $l$=0% | $l$=30% | $l$=0% |
| **SAGE** | 0.41 | 0.09 | 0.28 | 0.04 |

Table 6: Few-shot and Zero-shot generalization over technology and topology

In our experimental setup, we focused on utilizing a limited number of labeled samples to address the challenge of computational burden associated with generating labeled data. We considered a total of $n$ = 1k labeled samples and evaluated the performance based on the percentage of data ($l$) representing the infusion of technology/topology information for both few-shot and zero-shot learning. Given the constraints on labeled data availability, zero-shot learning was found to be ineffective in achieving satisfactory performance. However, there was some promising progress observed with few-shot learning, which indicates the potential of utilizing a small amount of labeled data to enhance the learning process.

## 7   Conclusion

Our new method for creating graph of circuits produced a consistent, homogeneous graph that can be used across multiple circuits. This approach lead to an accurate surrogate model with a computationally inexpensive node-level regression problem. By utilizing semi-supervised learning algorithms, we significantly reduced reliance on labeled datasets. The proposed GCX(GNN) surrogate model was further incorporated into EASCO and ASTROG. For the above mentioned test cases these algorithms outperformed the competing state-of-the-art optimization methods and yielded the most optimal design parameters that precisely meets the designer's criteria.

## 8   Acknowledgements

We extend our gratitude to the reviewers for their valuable comments and insightful suggestions, which have significantly enhanced the quality of this paper. We also appreciate the contributions of the entire MISN lab community for their engaging discussions related to this work. Our research has been made possible through the support of the DST Inspire faculty grant MI02322G.

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
