# OpenReview forum: "Graph of Circuits with GNN for Exploring the Optimal Design Space"
_NeurIPS.cc/2023/Conference — NeurIPS 2023 poster_

### Official Review · Reviewer_5fbs · 2023-07-05

**Soundness:** 2 fair
**Presentation:** 2 fair
**Contribution:** 2 fair
**Rating:** 3
**Confidence:** 4

**Summary:**

Automatic circuit design and optimization is an active field of research. Many different algorithms have been proposed with different design objectives, e.g., layout optimization, size optimization, and topology generation. Prior works adopted GNNs to represent circuit topologies, but the authors suggest a different approach in this work. After constructing a dataset consisting of sample circuits and their labels, they are mapped as a node in a graph. Then, a GNN model is employed to predict the labels of unlabeled samples. Two different optimization algorithms are proposed, and they exhibited relatively good performance in experiments.

**Strengths:**

- This is a new approach to applying GNNs to circuit design.
- The method can be applied to a wide range of circuit topologies (in theory).
- The surrogate model works relatively well in experiments.


**Weaknesses:**

- Probably the most interesting idea in this work is how to use graphs for circuit optimization. The way of embedding circuits into a node of a graph looks novel. However, the performance of the surrogate model is quite low, as displayed in Tables 1 and 2. It might be slightly better than prior methods, but its accuracy is still too low, so the method is not very practical. Can this really substitute conventional SPICE simulations when optimizing the design? There is no guarantee that this algorithm can produce an equally good circuit compared to SPICE-based approaches.
- Since the GNN models and optimization algorithms are not entirely new, this work might not see much interest in the machine learning community, especially considering prior works have been mostly published in EDA conferences and journals (e.g., DAC, ICCAD, and TCAD).


**Questions:**

- The optimized two-stage OTAs display good performances in Table 4. However, analog circuits often show totally different behavior in transient simulations. Were all the final designs verified in transient simulations?
- 2-stage OTA and ring oscillator are very simple circuits. Did the algorithm work well for more complex circuits?


**Limitations:**

- Not a practical substitute for SPICE models
- Only tested on simple circuits

---

> ### Author Rebuttal · Authors · 2023-08-09
>
> **W1, L1**: Thanks for your comments.
>
> **Accuracy of the surrogate model**:
> The model under consideration exhibits somewhat modest performance in metrics such as GAIN (0.6) and UGB (0.6). However, a contrasting trend emerges when evaluating metrics like GM, PM, Noise, Power, Frequency, and Delay, where the $R^2$ scores consistently surpass 0.8. This performance has been achieved under a very limited labeled scenario (comprising merely 500 labeled samples). This dataset size stands in stark contrast to the larger datasets employed by comparable studies.
>
> It's worth noting that the potential for performance enhancement remains by introducing a small number of additional samples. We also intend to explore encoding of the design space to further improve the $R^2$ score.
>
> Hence we believe that the proposed surrogate model can act as an effective proxy to the SPICE simulations during optimization.
>
> **Substituting SPICE simulations during optimization**:
>  The pre-trained surrogate model was employed during optimization and we performed an actual (through simulation) vs predicted value comparison corresponding to the most optimal design parameter's performance metrics. We observed that the predicted results from the surrogate model closely resembles with the simulated values across most metrics. Thus providing strong evidence that the proposed model can act as a proxy to SPICE simulations.
>
> **W2**: Thanks for your comments. Though graph has been introduced in the circuit designing problem this work is first to incorporate semi-supervised learning framework. This significantly reduces the reliance over extensive labeled datasets. Furthermore, our work yields the following benefits:
>
> (1) Formulating a node regression problem  by considering each node to be a circuit.
>
> (2) Incorporate technology and topology information to facilitate knowledge transfer.
>
> (3) Since the graph obtained is homogeneous it allows any GNN architecture to be easily integrated as opposed to prior works which generate heterogeneous graphs.
>
> (4) Incorporating semi-supervised learning framework, which has reduced the reliance on extensive labeled datasets.
>
> (5) Create a static and online framework incorporating graph-based surrogate models to obtain the most optimal parameters.
>
> **Focus to Machine Learning Community**:
> Though the algorithms underpinning our contributions draw upon existing knowledge but we provide a distinctive way of integrating them together and provide an efficient and accurate end-to end pipeline for the circuit designing problem.
>
> The integration of existing algorithms has become a highly intriguing and well-received area of research, capturing considerable attention in top-tier conferences like **NeurIPS** [2,3], **AAAI**[1], **ICLR** [6] and **ICML** [4,5] in **2022-23**. This integration is especially focused on circuit optimization, covering both analog and digital circuits.
>
> **References**:
> [1] Domain Knowledge-Based Automated Analog Circuit Design with Deep Reinforcement Learning, author={Weidong Cao and Mouhacine Benosman and Xuan Zhang and Rui Ma}, year={2022}.
>
> [2] Versatile Multi-stage Graph Neural Network for Circuit Representation, author = {Yang, Shuwen and Yang, Zhihao and Li, Dong and Zhang, Yingxueff and Zhang, Zhanguang and Song, Guojie and Hao, Jianye}, year = {2022}.
>
> [3] Learning to Design Circuits, author={Hanrui Wang and Jiacheng Yang and Hae-Seung Lee and Song Han}, year={2020}.
>
> [4] Learning to Design Analog Circuits to Meet Threshold Specifications,
>   author =       {Krylov, Dmitrii and Khajeh, Pooya and Ouyang, Junhan and Reeves, Thomas and Liu, Tongkai and Ajmal, Hiba and Aghasi, Hamidreza and Fox, Roy}, year = 	 {2023}
>
> [5]  Circuit-{GNN}: Graph Neural Networks for Distributed Circuit Design,
>   author =       {Zhang, Guo and He, Hao and Katabi, Dina}, year = {2019}
>
> [6] CktGNN: Circuit Graph Neural Network for Electronic Design Automation, author = {Zehao Dong and Weidong Cao and Muhan Zhang and Dacheng Tao and Yixin Chen and Xuan Zhang}, year = {2023}
>
> **Q1**: Thanks for asking this question and in fact this is a very important point. In this work we have simulated the design using optimized parameter obtained from our algorithm. The results shown in table.4 and table.5 in main paper as well as Table.1 and table.6 are generated from simulations only using the optimized parameter. Further, we have also performed a transient step response  of unity gain amplifier which uses our designed opamp. We observed from this step response that the system is stable and phase margin of this circuit should be somewhere between 50-55 degree (as there is only 1 visible peaking in the ringing response and then it settles) and thus it meet the desired performance. Kindly refer the simulation results supplemented.
>
> **Q2, L2**: Thanks for asking this question. Although two stage amplifier and ring oscillator are simple circuit but they cover a wide variety of circuits as two stage amplifier is an example of circuit which mostly works in linear region of operation where small signal analysis is valid and operating points are calculated only once while on the other hand ring oscillator operates mostly in large signal region where transistor operates in different region of operation and analysis are mostly performed using periodic steady states where operating points are continuously changing in a repetitive fashion. Further two stage OTA process voltage domain signal while in ring oscillator time domain information of signal is important. Therefore even though these circuits may look simple but it covers different and important category of analog circuits and covers a wide range of analog circuits. Most of the analog circuit can be thought of an extension and combination of these simple building blocks.

---

> > ### Comment · Reviewer_5fbs · 2023-08-17
> >
> > I appreciate the authors’ response. Unfortunately, many concerns have not been addressed yet, despite additional details provided in the rebuttal. I will not change the score.

---

> > > ### Author Response · Authors · 2023-08-17
> > >
> > > Thanks for your response. To the best of our understanding, we have tried to answer your concerns. We will do our best to address your remaining concerns but it will be very helpful if you could point us to the issues that need to be better resolved.

---

### Official Review · Reviewer_ufxr · 2023-07-06

**Soundness:** 3 good
**Presentation:** 4 excellent
**Contribution:** 3 good
**Rating:** 7
**Confidence:** 1

**Summary:**

- This paper propose a GNN based framework, dubbed as Graph of Circuits Explorer (GCX), to optimize circuit design by predicting the performance parameters of the nodes in circuits, e.g., Gain, Bandwidth, Noise, etc.
- This paper 1) utilizes a semi-supervised learning framework is employed for the graph-based surrogate model, and 2) creates a comprehensive feature vector that integrates information about various technology nodes and topologies, emphasizing generalizability in zero-shot and few-shot learning frameworks.
- With the proposed techniques, this work can enhance circuit performance prediction accuracy and efficiency while reducing the reliance on extensive labelled datasets.
- It demonstrated the proposed frameworks's effectiveness via simulation under 180nm CMOS technology, with model generalizability tested on 65nm and 45nm CMOS process nodes.

**Strengths:**

- The presentation of this paper is very solid and clear, both in terms of the text description and the figure illustration.
- This work conduct comprehensive validation across different GNN model variants, other circuit performance prediction methods, and different ablation settings. The improvements show are consistent.

**Weaknesses:**

- The contribution and distinction over the inherent characteristics of GNNs can be stressed out more. This paper extensively leverages the GNN's ability of predicting the unknown nodes' results based on their surrounding neighbors. However, this part has been widely researched for in the transductive learning scenario of GNNs and the label propagation in earlier graph processing works. The authors could better stress out these inherent (vanilla) techniques' limitations and what they do to specifically address them.
- Practical significance of the results' improvements could be supplemented. Although the evaluations consistently generate improvements, it is a little unclear what the technique can enable in the practical EDA tasks while the existing methods can be lacking. Also, the tested metrics' significance in actual applications can be added.
- Is there any potential limitation of the proposed methods?

**Questions:**

Please address the comments in the weakness section.

**Limitations:**

The authors have not yet included the limitations and broader impacts.

---

> ### Author Rebuttal · Authors · 2023-08-09
>
> **W1**: Thanks for the feedback. We had performed following set of experiments highlighting the robustness of our proposed algorithms to some of the commonly encountered problems with vanilla GNN architectures:
>
> (1)**Over-smoothing in GNNs**: Over-smoothing is an inherent problem with deeper GNNs and result in performance degradation.
>
> **Proposed solution**: (1) Generate sparse graphs: the formulation discussed in the paper provides sparsity control on the graph. As a result over-smoothing is controlled. (2) Use variants of GNN: GCX(SAGE) as proposed in the paper provides highest resilience to over-smoothing. (refer table for experimental results)
>
> | GCX(.) | Gain ($p=$50%) | Gain ($p=$70%) | UGB ($p=$50%) | UGB ($p=$70%) | GM ($p=$50%) | GM ($p=$70%) | PM ($p=$50%) | PM ($p=$70%) | Noise ($p=$50%) | Noise ($p=$70%) | Power ($p=$50%) | Power ($p=$70%) |
> |-------|------------|------------|-----------|-----------|----------|----------|----------|----------|-------------|-------------|-------------|-------------|
> | SAGE  | 0.33       | 0.58       | 0.30      | 0.55      | 0.76     | 0.81     | 0.65     | 0.79     | 0.73        | 0.93        | 0.90        | 0.92        |
> | GAT   | 0.07       | 0.11       | 0.05      | 0.06      | 0.64     | 0.78     | 0.63     | 0.66     | 0.32        | 0.40        | 0.50        | 0.51        |
> | GCN   | 0.03       | 0.09       | -0.05     | -0.01     | 0.09     | 0.33     | 0.13     | 0.13     | 0.08        | 0.10        | 0.40        | 0.45        |
>
>
> **Table 1: $R^2$ scores with different GNN architectures with deeper layers - Sparse Graph: (Average Degree- 3.5)**
>
> | GCX(.) | Gain ($p=$50%) | Gain ($p=$70%) | UGB ($p=$50%) | UGB ($p=$70%) | GM ($p=$50%) | GM ($p=$70%) | PM ($p=$50%) | PM ($p=$70%) | Noise ($p=$50%) | Noise ($p=$70%) | Power ($p=$50%) | Power ($p=$70%) |
> |-------|------------|------------|-----------|-----------|----------|----------|----------|----------|-------------|-------------|-------------|-------------|
> | SAGE  | 0.36       | 0.52       | 0.44      | 0.49      | 0.36     | 0.70     | 0.78     | 0.83     | 0.82        | 0.88        | 0.79        | 0.89        |
> | GAT   | 0.00       | 0.00       | 0.00      | 0.00      | 0.28     | 0.52     | 0.44     | 0.52     | 0.12        | 0.18        | 0.38        | 0.40        |
>
> **Table 2: $R^2$ scores with different GNN architectures with deeper layers - Dense Graph: (Average Degree- 10.1)**
>
> (2) **Rigidity to specific topologies**: Incorporating multiple topologies to facilitate generalizabilty has not been widely explored.
>
>    **Proposed solution**: We attempt to perform knowledge transfer across different technology files and topologies. We observe encouraging results for test case under study (refer the following table).
>
> | Model GCX(SAGE) | 3-S,FS (65,45nm) $p$=350 | 3-S,FS (65,45nm) $p$=450 | 3to5-S,FS (65,45nm) $p$=350 | 3to5-S,FS (65,45nm) $p$=450 |
> |----------------|------------------|------------------|-------------------|-------------------|
> | **Frequency**  | 0.73             | 0.86             | 0.45              | 0.78              |
> | **Delay**      | 0.76             | 0.85             | 0.66              | 0.72              |
> | **Power**      | 0.81             | 0.88             | 0.67              | 0.80              |
> | **Average**    | **0.77**         | **0.86**         | **0.59**          | **0.76**          |
>
> **W2**:
> **Practical Significance**:
> Minimize the dependence on extensive labeled datasets, consequently leading to a reduction in the duration of the design cycle.
>
> **Comparison with practical EDA tools**: (1) Current EDA tools completely leverage supervised learning framework which required extensive simulations. This makes the process resource and time intensive. Our proposed method employs semi-supervised learning to act as proxy to simulations under limited label scenario. (2) The usage of graph and graph-based surrogate models have not been widely explored in the EDA techniques and current works are very restrictive in terms of generalizability.
>
> **Significance to Applications**: We consider different metrics like Power consumption, Noise, PM etc which contribute to stability and efficient performance of the circuit. The demonstrated results validate that our proposed algorithm balances the trade-off between performance (metrics like Gain, UGB, Noise, PM) and resource consumption (power).
>
> **W3**: After thorough experimentation across various paradigms we observe our proposed approach has following limitations:
>
> **Low $R^2$ score**: We currently observe that $R^2$ score corresponding to metrics like Gain and UGB are low, which results in incorrect predictions. We intent to explore techniques like encoding of the design space to further elevate the score.
>
> **Limited success in knowledge transfer**: Our current approach shows promising results in facilitating knowledge transfer. However, to realize a more comprehensive replacement of SPICE simulations, it is imperative to enhance the implementation of our surrogate model to ensure its accuracy and reliability.

---

### Official Review · Reviewer_LaWa · 2023-07-07

**Soundness:** 3 good
**Presentation:** 3 good
**Contribution:** 3 good
**Rating:** 6
**Confidence:** 1

**Summary:**

This paper proposes a graph based semi supervised framework that is capable of driving analog design

**Strengths:**

* The paper is generally well written and organized.
* The figures are nicely drawn to make understanding necessary concept easier.
* The proposed mehtod is well illustrated, the idea of using semi-supervised methods looks interesting and promising.
*  Experimental results are well demonstrated and explained, comparison against existing methods look promising.

**Weaknesses:**

Please see questions.

**Questions:**

* on conclusion, from what I understand the result is more optimal than other methods, how to determine that it yields the most optimal parameters?
* For different technologies, does it need separate training or the knowledge is transferrable across technoilogies?

**Limitations:**

* The authors discuss some of the limitations in knowledge transfer.

---

> ### Author Rebuttal · Authors · 2023-08-09
>
> **Q1**: This is a very important question and thanks for pointing it out.
>
> **Algorithmic viewpoint**: Optimality of the design parameters depends on how well the objective function is optimized. For the test cases we considered, FOM was chosen as the objective function (details are discussed in supplementary). When the stopping criteria (discussed in Q3- reviewer LVTs) is met the algorithm gives out the most optimal set of design parameters.
>
> **Designer's viewpoint**: The significance assigned to each metric transforms across different circuits and applications. For an amplifier, gain might reign supreme, whereas Phase Margin (PM) might command precedence elsewhere. Conversely, lower power consumption could be paramount in other scenarios. In each case the definition of optimality changes and results in different set of design parameters. For our case, after taking the designer's recommendation we assign equal importance to all the specifications under consideration and the algorithm provides the optimal parameters. Notably, our algorithm can easily incorporate these additional attributes and give out the parameters accordingly.
>
> **Q2**: Thanks for asking this question. As discussed in the paper we attempted to perform few-shot and zero-shot learning by generating a comprehensive feature vector which facilitates knowledge transfer across technology nodes and topology. Our proposed approach achieved an encouraging performance in few-shot learning and the results are as follows:
> | Model GCX(SAGE) | 3-S,FS (65,45nm) $p$=350 | 3-S,FS (65,45nm) $p$=450 | 3to5-S,FS (65,45nm) $p$=350 | 3to5-S,FS (65,45nm) $p$=450 |
> |----------------|------------------|------------------|-------------------|-------------------|
> | **Frequency**  | 0.73             | 0.86             | 0.45              | 0.78              |
> | **Delay**      | 0.76             | 0.85             | 0.66              | 0.72              |
> | **Power**      | 0.81             | 0.88             | 0.67              | 0.80              |
> | **Average**    | **0.77**         | **0.86**         | **0.59**          | **0.76**          |
>
>  So, by fine-tuning our pre-trained model to the newer set of samples our approach facilitates knowledge transfer.
>
> **L1**: After thorough experimentation across various paradigms we observe our proposed approach has following limitations:
>
> (1) **Low $R^2$ score**: We currently observe that $R^2$ score corresponding to metrics like Gain and UGB are low, which results in incorrect predictions. We intent to explore techniques like encoding of the design space to further elevate the score.
>
> (2) **Limited success in knowledge transfer**: Our current approach shows promising results in facilitating knowledge transfer. However, to realize a more comprehensive replacement of SPICE simulations, it is imperative to enhance the implementation of our surrogate model to ensure its accuracy and reliability.

---

### Official Review · Reviewer_pjdD · 2023-07-20

**Soundness:** 3 good
**Presentation:** 3 good
**Contribution:** 3 good
**Rating:** 6
**Confidence:** 3

**Summary:**

This article proposes a graph-based circuit design framework to help address the issue of label scarcity in circuit design. The paper is well-written, virtually free of errors or inconsistencies. Information is presented accurately and timely, creating a smooth narrative flow between the main paper and supplementary content. However, I do have some queries regarding the specific technical aspects of the paper. I would be immensely grateful if the authors could provide satisfying answers.

**Strengths:**

(1) The technology presented is quite innovative, successfully merging popular Graph Representation Learning with the semi-supervised nature of GNNs and the fundamental characteristics of circuits.
(2) The writing exhibits smooth logic, with technical details clearly expressed.

**Weaknesses:**

(1) The citations in this paper primarily focus on circuit design and system-related conferences or journals. I would like to see more analysis related to GNNs, as GNNs typically require deeper layers for larger circuits (components of which should be quite complex). However, an overly deep receptive field might lead to an over-smoothing issue. Is there a chance of over-smoothing occurring when each device is learning within the network?

(2) While this paper presents many circuit-related simulation experiments, it lacks validation in real-world scenarios. To my knowledge, circuit design in practical settings is costly, which may present challenges for the authors in testing on specific circuits. Nevertheless, I am still interested in querying this issue and hope the authors could offer some related thoughts.

(3) The importance of each device in the circuit might vary. In certain situations, some key devices are crucial to the entire circuit. Therefore, I believe the authors' discrete encoding approach may be unsuitable.

**Questions:**

 (1) Please include more detailed analysis of the specific GNN networks, especially addressing potential issues such as over-smoothing.

(2) Is it feasible to conduct experiments in real circuits? If so, please include such tests, which could involve smaller, simpler circuits. If not, it would be helpful to clearly explain the reasons for not testing in real-world situations.

**Limitations:**

(1) Please include tests on various GNN frameworks.

(2) Please clarify in the experiment why only simulation results are presented.

---

> ### Author Rebuttal · Authors · 2023-08-09
>
> **W1, Q1, L1**: Thanks for raising the concern. We experimented by creating deeper GNNs and the results are as follows:
> | GCX(.)    | Gain ($p$=50%) | Gain   ($p$=70%) | UGB ($p$=50%) | UGB ($p$=70%) | GM ($p$=50%) | GM ($p$=70%) | PM ($p$=50%) | PM ($p$=70%) | Noise ($p$=50%) | Noise ($p$=70%) | Power ($p$=50%) | Power ($p$=70%) |
> |-----------|----------------|----------------|---------------|---------------|--------------|--------------|--------------|--------------|----------------|----------------|----------------|----------------|
> | SAGE      | 0.33           | 0.58           | 0.30          | 0.55          | 0.76         | 0.81         | 0.65         | 0.79         | 0.73           | 0.93           | 0.90           | 0.92           |
> | GAT       | 0.07           | 0.11           | 0.05          | 0.06          | 0.64         | 0.78         | 0.63         | 0.66         | 0.32           | 0.40           | 0.50           | 0.51           |
> | GCN       | 0.03           | 0.09           | -0.05         | -0.01         | 0.09         | 0.33         | 0.13         | 0.13         | 0.08           | 0.10           | 0.40           | 0.45           |
>
> **Table 1: $R^2$ scores with different GNN architectures with deeper layers - Sparse Graph: (Average Degree- 3.5)**
> | GCX(.)    | Gain ($p$=50%) | Gain ($p$=70%) | UGB ($p$=50%) | UGB ($p$=70%) | GM ($p$=50%) | GM ($p$=70%) | PM ($p$=50%) | PM ($p$=70%) | Noise ($p$=50%) | Noise ($p$=70%) | Power ($p$=50%) | Power ($p$=70%) |
> |-----------|----------------|----------------|---------------|---------------|--------------|--------------|--------------|--------------|----------------|----------------|----------------|----------------|
> | SAGE      | 0.36           | 0.52           | 0.44          | 0.49          | 0.36         | 0.70         | 0.78         | 0.83         | 0.82           | 0.88           | 0.79           | 0.89           |
> | GAT       | 0.00           | 0.00           | 0.00          | 0.00          | 0.28         | 0.52         | 0.44         | 0.52         | 0.12           | 0.18           | 0.38           | 0.40           |
>
> **Table 2: $R^2$ scores with different GNN architectures with deeper layers - Dense Graph: (Average Degree- 10.1)**
>
> GCN exhibits worst performance when subjected to deeper layers. Initial experiment (best performing model) was conducted with GCN (2 Hidden layers), GraphSAGE (1 Hidden layer) and GAT (1 Hidden layer). To demonstrate over-smoothing we incorporate additional layer to the corresponding architectures.
>
> We have acknowledged the occurrence of "over-smoothing" as GNNs become deeper, potentially compromising surrogate model accuracy. To counter this, we've adopted strategic measures during algorithm development to curb over-smoothing's adverse effects effectively:
>
> **Construct Sparse graph**- The optimization formulation suggested in the paper controls the sparsity of the graph, thus reducing the effect of over-smoothing.
>
> **Use variants of GNNs**: Architectures such as SAGE and GAT exhibit enhanced resilience to over-smoothing. Our proposed GCX(SAGE) model, showcases superior performance in mitigating the over-smoothing challenge.
> By adopting the above mentioned preventive techniques we obtain significant resilience to the over-smoothing problem with deeper GNNs.
>
> **W2, Q2, L2**:Thanks for asking this question. It is true that performance of circuit design is finally tested on a silicon. However, process of designing, fabricating it in chip and validating in lab is a costly and time taking affair. Hence these designs are first validated through extensive simulation to ensure the performance. As the model used in these simulations are obtained from real validation on silicon hence they closely predict the behaviour of circuits realized on silicon. Currently we have also simulated all the optimized circuits using the same models (180nm CMOS process TSMC foundry models which are made available by foundry) to ensure its correctness and our results are shown in supplementary material. We have a plan to validate it on silicon however this process will require another year as we are currently in design process and fabrication and getting chip back will take at least an year or so. Hence we at this stage are providing simulation results which in circuit community is a close confirmation of validity of a circuit.
>
> **W3**: Thanks for the comments. Our present encoding technique efficiently consolidates crucial circuit information into a singular feature vector. Nonetheless, it remains evident that specific components within a circuit might hold greater significance than others. While our current encoding methodology assigns uniform weightage to all components, our approach offers the advantage of tailoring weightage to specific design parameters. This adaptability enhances the comprehensibility of feature encoding and enables a finer grasp of the overall circuit configuration.

---

### Official Review · Reviewer_LVTs · 2023-07-29

**Soundness:** 3 good
**Presentation:** 2 fair
**Contribution:** 2 fair
**Rating:** 5
**Confidence:** 3

**Summary:**

The design automation of analog circuits poses significant challenges in terms of the large design space, complex interdependencies between circuit specifications, and resource-intensive simulations. To address these challenges, this paper presents an innovative framework called the Graph of Circuits Explorer. GCX enables the creation of a surrogate model that facilitates efficient exploration of the optimal design space within a semi-supervised learning framework which reduces the need for large labelled datasets. The proposed approach comprises three key stages. The effectiveness of the proposed approach is demonstrated through simulated performance evaluation of various circuits, using derived parameters in 180 nm CMOS technology. Furthermore, the generalizability of the approach is extended to higher-order topologies and different technology nodes such as 65 nm and 45 nm CMOS process nodes.

**Strengths:**

- This paper introduces a novel and innovative approach to optimize circuit design by leveraging graph representation and graph-based semi-supervised learning. The approach is designed to enhance accuracy and efficiency while reducing the reliance on extensive labelled datasets.
- To achieve this goal, a semi-supervised learning framework is employed for the graph-based surrogate model. a new method is introduced to create a comprehensive feature vector that integrates information about various technology nodes and topologies, emphasizing generalizability in zero-shot and few-shot learning frameworks. By integrating these approaches, two new optimization methods on graph-based surrogate models are proposed: Efficient Analog Sizing via Constrained Optimization (EASCO) and Analog Sizing through Real-time Online Graphs (ASTROG).


**Weaknesses:**

- This paper leverage a GNN,  more experiences about how to extract the feature need to be performed, such as transformer
- The process of training is not claimed clearly, the paper claims that "Semi-Supervised Learning on Graph of Circuits", More details need to be added. such as the process of Semi-Supervised Learning.
- How to define the node weights of a graph? when training process, does the relation between two nodes is neccessary for training?
-"Stopping criteria are predefined, and the optimal design is outputted if these criteria are met; otherwise, the process moves to the next step.", the definition of "Stopping criteria" is not claimed clearly.
- the details of dataset is not claimed clearly.
- "Given the constraints on labelled data availability, zero-shot learning was found to be ineffective in achieving satisfactory performance. However, there was some promising progress observed with few-shot learning, which indicates the potential of utilizing a small amount of labelled data to enhance the learning process." more details and analysis need to be provided

**Questions:**

as written in Weaknesses

**Limitations:**

as written in Weaknesses

---

> ### Author Rebuttal · Authors · 2023-08-09
>
> **W1**: Thanks for the feedback. We tried experimenting with Graph Transformer architecture across different performance metrics with appropriate hyper-parameter tuning. The loss function shows a swift decline  however the obtained $R^2$ scores are very poor.
> With our current time constraints in mind, our focus remains on investigating the implementation details to better understand the underlying factors influencing our outcomes.
>
> **W2**: Thanks for raising the concern. We hope the following discussion provides a more detailed description:
> (1) Start by generating a detailed dataset (details described in W4) and $l$ labels are simulated to corresponding samples, where $l << n$. (2) Subsequent step is creation of a graph comprised of circuit instances, denoted as $X=(X_l, X_u)$; represented as a weighted adjacency matrix ($W$).(3) Within this graph, each circuit instance $i$ corresponds to a node. Corresponding circuit-level parameters form a feature vector $(X_i \in \mathbb{R}^{k})$.(4) Termed as the Graph of Circuits Explorer (GCX), this graph, $G(V,E,W,X_u,X_l,Y_l)$, is seamlessly integrated with Graph Neural Networks (GNNs) to facilitate label propagation across unlabeled samples. (5) Semi-supervised learning framework of GNNs accurately learn the underlying graph function while constraining loss computation solely to the available labeled samples. (6) We employ label propagation—on performance metrics such as Gain, UGB, GM,  PM etc—across unlabeled circuit instances ($|X_u|$-nodes).(7) We have  two distinct labeled data scenarios, specifically $p= 30\\%$ and $p=50\\%$ , for training the Graph Neural Networks (GNNs).
>
>  **W3**: **Node weights in a graph**: Thanks for raising the concern. The weights assigned to nodes in a graph are obtained based on the proximity of design parameters among different circuit instances. When the Euclidean distance between these design parameters is minimal, the corresponding instances are assigned higher edge weights.  This leads to more pronounced connections between circuit instances that share similar design parameters within a $d$-dimensional space, in contrast to  instances characterized by distinct parameters. The resulting node weights are derived as a weighted adjacency matrix, achieved through the resolution of an optimization formulation outlined in the paper.
>
> **Significance of node weights**:
> (1) **Sparsity in graph**- Computational time reduces during training, over-smoothing problem can be controlled.
> (2)**Identifying prominent neighbors**- Edge weights help to get rid of unnecessary connection of the node. Information via message passing is obtained only through important neighbors.
>
> **Stopping criteria**: For both the test cases we exclusively define our objective function as Figure of Merit (FOM). Details about FOM can be found in the supplementary material.
> Stopping criteria is achieved when optimum value of FOM is reached i.e $[ (FOM)_n - (FOM)_{n-1} ] \leq \epsilon$ where $n$ is the number of iterations  in both EASCO and ASTROG algorithms. Each algorithm is run TEN times to ensure reproducibility (specifications are satisfied).
>
> **W4**: Thanks for raising the concern.  Design space bounds, Performance metrics and Specifications are carefully considered after designer's recommendation (mentioned in experiment section). Features (design space points) $X = (X_u, X_l)$ are generated by uniformly sampling points across the $d$-dimensional space forming a matrix of size $X \in \mathbb{R}^{n \times d}$ . Since we employ semi-supervised learning we simulate  labels corresponding to only $l$ samples where $l << n$ (samples are randomly chosen to avoid bias). The dataset obtained is : $(X_l,X_u,Y_l)$. Refer table for better understanding:
>
> | Samples | Design parameters | Performance metrics |
> |---------|-------------------|--------------------|
> | 1       | $X_1$             | $Y_1$              |
> | 2       | $X_2$             | $Y_2$              |
> | -       | -                 | -                  |
> | $l$     | $X_l$             | $Y_l$              |
> | $l+1$   | $X_{l+1}$         | $\times$           |
> | -       | -                 | -                  |
> | $n$     | $X_n$             | $\times$           |
> **Sample Distribution**: $X_{1-l}, Y_{1-l}$ corresponds to labeled samples; Samples from $X_{(l+1)-n}$ constitute $X_u$ unlabeled samples.
> Two-stage Miller Compensated OTA: Design parameters ($X_i$) are: Reference current ($I_{ref}$), Reference Voltage ($V_{dd}$), Transistor widths and length ($W_{1-8}, L_{1-8})$, Capacitance ($C_c,C_L$) and load resistance ($r_L)$. Performance metrics ($Y_i$) are : Gain, UGB, GM,PM, Noise, Power.
>
> Three-stage Ring Oscillator: Design parameters ($X_i$) are: Reference Voltage ($V_{dd}$), Transistor widths and length ($W_{1,2}, L_{1,2})$ and Capacitance ($C_1$). Performance metrics ($Y_i$) are: Frequency, rms Jitter, Delay, Power.
>
> **W5**: Thanks for raising the  concern. We performed two set of experiments with $p$= 350 and  450 labeled samples for technology nodes $65\,nm$ and $45\,nm$ with $180\,nm$
> Technology transfer: $180\,nm \rightarrow 65\,nm$ and $45\, nm$ for 3-stage.
> Technology + Topology transfer: $180\,nm \rightarrow 65\,nm$ and $45\, nm$ from 3-stage to 5-stage.
> | Model GCX(SAGE) | 3-S,FS (65,45nm) $p$=350 | 3-S,FS (65,45nm) $p$=450 | 3to5-S,FS (65,45nm) $p$=350 | 3to5-S,FS (65,45nm) $p$=450 |
> |----------------|------------------|------------------|-------------------|-------------------|
> | **Frequency**  | 0.73             | 0.86             | 0.45              | 0.78              |
> | **Delay**      | 0.76             | 0.85             | 0.66              | 0.72              |
> | **Power**      | 0.81             | 0.88             | 0.67              | 0.80              |
> | **Average**    | **0.77**         | **0.86**         | **0.59**          | **0.76**          |

---

### Author Rebuttal · Authors · 2023-08-10

**Q1)** **Reviewer-5fbs**: Transient Analysis Simulation results

---

### Decision · Program_Chairs · 2023-09-21

**Decision:**

Accept (poster)

**Comment:**

This paper applies graph neural networks to circuit design and optimization. 4 reviewers are positive about the work, and 1 reviewer has a negative opinion after the rebuttal. Most reviewers have relatively low confidence because this paper is not a very common topic in machine learning conferences, but the authors have pointed out that there have been several papers on this topic in the past two years in NeurIPS/AAAI/ICML. The authors are recommended to add the discussion on SPICE simulations and surrogate models to the paper. Overall, a new and interesting approach is proposed to circuit design optimization with satisfactory performance. Thus, I would recommend acceptance of this paper.